# Anti-*Toxoplasma gondii* IgM Long Persistence: What Are the Underlying Mechanisms?

**DOI:** 10.3390/microorganisms10081659

**Published:** 2022-08-17

**Authors:** José Antonio Vargas-Villavicencio, Irma Cañedo-Solares, Dolores Correa

**Affiliations:** 1Laboratorio de Inmunología Experimental, Instituto Nacional de Pediatría, Mexico City 04530, Mexico; 2Dirección de Investigación/Centro de Investigación en Ciencias de la Salud, FCS, Universidad Anáhuac México Campus Norte, Av Universidad Anáhuc 46, Lomas Anáhuac, Huixquilucan 52786, Mexico

**Keywords:** *Toxoplasma gondii*, IgM, serological diagnosis

## Abstract

Diagnosis of *Toxoplasma gondii* acute infection was first attempted by detection of specific IgM antibodies, as for other infectious diseases. However, it was noted that this immunoglobulin declines slowly and may last for months or even years. Apart from the diagnostic problem imposed on clinical management, this phenomenon called our attention due to the underlying phenomena that may be causing it. We performed a systematic comparison of reports studying IgM antibody kinetics, and the data from the papers were used to construct comparative plots and other graph types. It became clear that this phenomenon is quite generalized, and it may also occur in animals. Moreover, this is not a technical issue, although some tests make more evident the prolonged IgM decay than others. We further investigated biological reasons for its occurrence, i.e., infection dynamics (micro-reactivation–encystment, reinfection and reactivation), parasite strain relevance, as well as host innate, natural B cell responses and Ig class-switch problems inflicted by the parasite. The outcomes of these inquiries are presented and discussed herein.

## 1. Introduction

*Toxoplasma gondii* is an obligate intracellular parasite distributed globally in warm-blooded animals, including humans, with variable incidence depending on climatic conditions, being higher in tropical areas than in cold or dry regions [1]. The acquired infection is caused by the ingestion of tissue cysts present in undercooked meat or oocysts that contaminate water or food, and that come from an environment where felines-definitive hosts-release them with their feces (Figure 1) [2]. The congenital form is due to vertical transmission from mother to offspring during pregnancy [3,4]. In immunocompetent individuals, it is usually asymptomatic [5] because both the innate and the adaptive cellular and humoral immune responses are capable of systemically destroying most of the gut-invading stages-bradyzoites or sporozoites-and the fast-proliferating tachyzoites, some of which convert into the cyst-forming bradyzoites leading to the chronic phase (Figure 1) [6,7,8,9,10,11,12]. The CD4+ lymphocytes in the lymph nodes orchestrate the adaptative response, which stimulates protective actions of phagocytes, Natural Killer (NK) and CD8+ cells, as well as B lymphocytes that systematically synthesize IgM and then IgG antibodies [6,7,8,9,10,11,12]. The IgA antibodies have been related to the local, or mucosal, response of the digestive tract, while IgE antibodies are produced during the acute infection, but they persist in severe cases or in prolonged lymphadenopathy [10].

In some cases of acquired infection, clinical problems arise in the form of lymphadenopathy or retinochoroiditis; they have been related to an undesirable immune profile or to the acquisition of virulent strains, which prevail in tropical zones of the world. Remarkably, the immune response profile and the clinical outcome seem to depend on the parasite type acquired [13,14,15,16,17,18].

The rate of clinical problems in congenitally infected cases is inversely related to the time at which the mother is infected during gestation, which in turn is related to the immune response maturity/profile of the embryo/fetus, but also to the type of parasite and the maternal immune response [4,19,20,21,22,23].

Diagnosis of the acute infection has been one of the main research motivations worldwide; this is the period of danger for the host and of higher probability of vertical transmission; moreover, it is important in terms of treatment efficacy, at least with the drugs routinely used, since they mostly target the tachyzoite stage [24,25]. Although the search for IgM antibodies was attempted to identify the acute infection, as for other infectious diseases [26], it soon became clear that IgM may last for months or even years [27,28,29,30,31,32,33,34,35,36,37,38,39,40,41,42,43,44]. As a result, several other old approaches were used, and others emerged, including serial sample analysis, measurement of IgG avidity and detection of parasite DNA [33,45,46].

Like other groups, we struggled with this phenomenon because it made it difficult to diagnose the acute infection in pregnant women and to estimate the risk of vertical transmission for medical management and for clinical and epidemiological investigations [21,47,48,49]. However, beyond the diagnostic problems caused by this “chronic IgM” phenomenon, the biological explanation for its occurrence attracted our attention from a basic point of view, since, as it will be seen, it cannot be entirely explained by technical problems. We wanted to know whether the IgM decay kinetics is similar among different patient groups, i.e., pregnant women, patients with clinically active acquired infection and babies with the congenital form.

In this review, we present data on the generality of this phenomenon, the importance of technical issues and the possible biological mechanisms that can explain it.

## 2. Is the “Chronic” IgM Phenomenon Occurring in A Specific Patient Group?

The long duration of the anti-*T. gondii* IgM antibodies was observed in the 1980s [36,50], and it has repeatedly been observed ever since. Therefore, we looked for reports containing data on the duration of the IgM response and found some studies with pregnant women under screening programs or of clinical cases with acquired toxoplasmosis, either in groups -including a documented outbreak- or isolated cases [27,31,33,34,35,36,37,38,39,40,41,42,44,51,52,53,54,55]. There was only one study in which IgM antibodies were followed in congenitally infected children up to week 40 [30]. Some papers report the proportion of positives (or negatives), while others provide information about IgM levels (mean or median) (Figure 2 and Figure 3). Several studies of people with acquired infection report the duration of IgM for 40 to 50 weeks [31,33,36], and those with pregnant women for 25 to 65 [38,42,51,52], and even some in which the presence of IgM antibodies was followed beyond week 70 [35,37,40,44]. A rather heterogenic kinetics among patients with acquired infection can be seen, partially depending on the technique used (see two examples in Figure 4), but it can be said that the persistence of IgM has been reported in gestational and acquired infection; for congenitally infected children, the decay was faster than most acquired cases, although some of them remained positive for as long as 30 weeks after birth (Figure 2) [30].

## 3. Specificity/Sensitivity Problems of Serological Tests

Various serological methods have been used for the detection of IgM antibodies, and it can be observed from the graphs of Figure 2 and Figure 3 that there are notable differences among them; in addition, there are differences according to presentation of the results, i.e., proportion of positive cases versus antibody levels (Figure 2 and Figure 3, respectively). To enhance these facts, we plotted the work by Takahashi and Rossi [35] in the right panel of Figure 4; they used samples from individuals with acquired infection and found discrepancies in both aspects, so they obtained a duration from 39 to more than 80 weeks (Figure 4) [35]. In the study by Del Bono et al. [37], using three different tests, IgM remained positive in some patients for periods of 70 weeks or longer (Figure 4, left panel) [37]. Finally, Fricker-Hidalgo et al. [51] reported curves of IgM detected by ISAGA, IFI, Vidas and Architect Toxo^®^. The levels of IgM reached a peak approximately three, four and five weeks after infection, respectively. With the ISAGA and Vidas tests, IgM antibodies remained positive longer than week 25. In contrast, IgM IFAT titers negativized around 13 weeks after infection. Using Architect Toxo^®^, the IgM levels were relatively higher than those with the Vidas test in the first four months, but at a low titer until week 24 (compare the lines labelled with diamonds in the left-upper panel of Figure 3 with those in the left-lower panel).

Based on these articles, it can be concluded that the method is important and that highly sensitive methods detect IgM for a longer time or are less specific. In some works, in which IgM levels are reported using methods such as ISAGA, chemiluminescence and ELISA, high titers are also observed for prolonged times. In contrast, IFAT shows high titers that rapidly decrease after week 7; in some cases, it becomes negative at week 13, although in others, titers of 10% are detected up to week 25 post-infection. 

These discrepancies rely on the properties of each technique, i.e., its diagnostic performance, the antigen employed in the assay and the cut-off point. Some authors are of the opinion that the kits for the detection of anti-*Toxoplasma* IgM antibodies are generally set up for screening, so they are highly sensitive but poorly specific [35], and that the use of tests with low specificity have led to the misdiagnosis of acute toxoplasmosis [56].

Despite the variability of the methods, it has been possible to demonstrate the persistence of IgM antibodies at least in a proportion of cases, and this is supported by single-case reports, such as that of an adult male who presented *T. gondii* infection confirmed by serology, clinical signs and persistent IgM for up to 7 years, while he presented low-avidity IgG antibodies for 10 months (Figure 3) [27]. In a second case of systemic toxoplasmosis affecting a protein S-deficient patient, there was even an increase in IgM antibody levels that remained positive at least 20 weeks after the onset of ocular symptoms (Figure 3) [29]. In addition, an individual with suspected toxoplasmosis as a cause of tubulointerstitial nephritis and uveitis syndrome after meningitis and systemic lymphadenopathy was shown to have IgM antibodies, but avidity results pointed to a chronic infection [55].

Therefore, based on the data presented in Figure 2, Figure 3 and Figure 4, and on many other studies that did not perform a kinetic analysis but reported that IgM for the diagnosis of acute infection was of low value, it can be concluded that the phenomenon of “chronic IgM” is universal. However, the reason for IgM persistence may vary among patients or groups.

An important difference between pregnant women and patients with acquired infection is that the former are mostly asymptomatic, since they are diagnosed by laboratory methods during screening programs, while all cases in the second group are followed after the appearance of signs and symptoms. This is relevant since it is the tachyzoite that causes the symptoms by invasion, replication and exocytosis, which produces necrosis of the host tissues. Since *T. gondii* is clinically silent in most infected persons [1,5], there are no data on the duration of the acute infection and of IgM antibodies in the largest infected population of the world. One might think that studies in pregnant women would give some insight into the response in asymptomatic people, but the immunological environment is special during most of the gestation period; also, hormones such as relaxin and progesterone regulate the Th1 and Th2 adaptive immune profiles [57,58,59]. In this regard, toxoplasmosis develops more frequently and with higher intensity in female than in male mice [60], and preliminary data of our group showed that pregnant mice present a greater parasitic load and more extended tissue lesions than non-pregnant ones [61]. It has been reported that estrogens induce cell cycle processes in the tachyzoite, which may influence its survival and the prolongation of the acute infection [62]. There are several studies indicating that low-avidity IgG antibodies can last for several weeks and even months during gestation [63,64,65,66], but the drug scheme is another important aspect to consider since pregnant women are provided with prophylaxis to avoid or diminish the risk of vertical transmission [24,40]. Interestingly, several independent works showed no differences in the permanence of IgM and of low-avidity IgG antibodies between women treated with spiramycine and those managed with pyrimethamine- sulfadiazine-folinic acid, the latter being the most effective scheme for destroying the tachyzoite [40,63,64,65,66]. Conversely, a significant difference in avidity kinetics was observed between congenitally infected children treated with the tri-therapy and those managed with spiramycine, showing delayed maturation of avidity in the first group [65].

The group of infants with congenital infection followed for IgM (Figure 2) was heterogeneous: some cases were suspected before delivery, others were detected through postnatal screening, and several arrived at the hospital already with clinical symptoms; thus, it was not discerned whether the kinetics was different between subclinical and diseased infants [30]. In any case, the decay of IgM antibodies in this group was shorter than for the other groups, although it was not a classical primary (days-long) immune response. An important aspect to consider for congenital infection is that they are infected directly through the blood during fetal life, while adults acquire it orally; this singularity may be important since more sources of IgM are possible in cases that respond to parasite antigens at the mucosae (see below).

In the following sections, we present possible explanations for the long duration of IgM antibodies in *T. gondii* infection.

## 4. Explanations of the Chronic *T. gondii*-IgM Phenomenon

### 4.1. Dynamics of the Infection: Antigenic Variation

The phenomenon of antigenic variation could be a cause of IgM persistence. This was described long ago in sleeping sickness caused by *Trypanosoma brucei* and in malaria due to the apicomplexan parasites that belong to the genus *Plasmodium* [67,68,69]. The hosts of these pathogens are continually exposed to new variants because they change the antigenic profile on the surface, inducing waves of immune response over months or even years. In malaria, antigenic variation has been directly related to the persistence of IgM antibodies [67]. Although there is no evidence that this phenomenon occurs in toxoplasmosis, there are reports of antigenic differences between the stages of the parasite, that is, the sporozoite, the tachyzoite and the bradyzoite, and the changes between two stages could induce IgM antibodies against new or “forgotten” antigens. For example, during sporozoite or bradyzoite to tachyzoite conversion or during encystment (tachyzoite to bradyzoite/tissue cyst transition), large changes occur in the transcriptome and, as a result, in the antigen array presented to the immune system [70,71]. All these changes would be of short duration, however, and should not elicit long-lasting primary IgM response. Other explanations are considered below.

### 4.2. Long-Lasting Acute Infection or Continuous Stimulation by Cysts?

Some evidence suggests that tachyzoites can persist for several months, mainly because the parasite DNA has been found in the blood of patients with chronic infections -as suggested by high avidity values- and vertical transmission occurrence [48]. However, there are some difficulties in sustaining this notion: as mentioned, the tachyzoite is highly active in replication and destruction of host cells and tissues, so the patients should have clinical problems while this parasitic stage persists. In support of this, Donadono et al. reported that pregnant women with toxoplasmosis and lymphadenopathy have a significantly higher risk (OR = 2.9; CI95%, 1.28–6.58) of transmitting the infection to their offspring than those who were asymptomatic [72]. Nevertheless, it is unclear from all data collected for this review that this is always true; most studies did not describe how long clinical problems persisted after treatment was started; therefore, it is unknown whether the proportion of patients who remained IgM-positive were those with unresolved disease (Figure 2, Figure 3 and Figure 4). Secondly, many studies in experimental models suggest that the tachyzoite is destroyed soon after the innate and adaptive immune responses are activated, and that some of them encyst and keep immunologically more silent [5]. There is also evidence in humans: as mentioned before, the duration of IgM antibodies in pregnant women who transmitted vs. those who did not transmit the infection to their offspring is not significantly different [40]. Likewise, Bertozzi et al. reported a case of acquired infection that presented a 7-year-long IgM duration despite the symptoms having disappeared 5 weeks after chemotherapy initiation [27].

There is a mirrored kinetics of IgM and avidity throughout infection, and a similar inverse correlation between these markers, regardless of the way the data are reported (Figure 5) [27,33,44,54,73]. Nevertheless, the kinetics in two of these studies showed that, despite reaching a steady “chronic” avidity value, IgM antibodies did not disappear, but prevailed at low or very low levels for several weeks or months (see upper and middle graphs of panel A in Figure 5). From these data gathered from original papers, we are uncertain that the avidity tests are really marking the acute (dangerous) infection. In this regard, children treated with pyrimethamine/sulfadiazine retained a low avidity of IgG antibodies for several weeks [65]; they may be experiencing the effect of the drug combination itself, which is aplastic, so it diminishes the production of all blood cells, including B-2 lymphocytes, which are those capable of class switching and antibody maturation (i.e., increase in affinity) [74].

As mentioned, *T. gondii* tachyzoites are controlled by the immune system, which keeps the parasite encysted for long periods [70]. Cysts could maintain a low-level stimulation of the immune system, and IgM antibodies could remain at low titers, but for a long time. Encystment is favored in infections with low or moderately virulent strains in more resistant hosts; interestingly, some reports in experimental infections or xenodiagnosis attempts support this notion. Mouse models using virulent (Ckn2) and non-virulent (Me49) strains revealed greater IgM reactivity against the virulent ones during the first 15 days after infection; however, the slope of IgM decline was steeper compared to that of the infection with the avirulent strain [75] (Figure 6). Unfortunately, IgM follow-up in animals is scarce and normally lasts for up to 60–70 days post-infection. There is heterogeneity in the inoculation via parasite strain and dose employed. Most articles found that the peak of anti-toxoplasma IgM was in the second week, with a decrease in the fourth, as for many other infectious diseases [75,76,77,78]. This response remained positive for up to the ninth week in some cases challenged with low-virulence strains, although the experiments were not continued, so it was unclear whether the response could last longer [76,78,79].

### 4.3. Reactivation/De-Encystment

It has been described in the literature that immunosuppression in people who harbor a chronic infection may cause excystment of *T. gondii* bradyzoites that convert back into tachyzoites [70], provoking a new acute infection that may stimulate B cells to produce antibodies against “forgotten” antigens. This has been demonstrated in patients with HIV who develop “Toxoplasmic Encephalitis”, but these antibodies are of the IgG isotype, not IgM [80,81]. Likewise, some cases of *T. gondii*-infected transplant recipients treated with immunosuppressive therapy before and after transplantation produce IgM antibodies when the infection reactivates, although this is not frequent, with IgG antibodies being the main reactivation response [82,83]. Finally, in another case of systemic toxoplasmosis involving a female patient with protein S deficiency, there was an increase in IgM antibody levels during the first 20 weeks after the onset of ocular symptoms, which supports a reactivation of the latent disease [29].

Sibalic et al. reported one child diagnosed with congenital toxoplasmosis in the first year of life who developed ocular toxoplasmosis 10 years later and presented high titers of IgM in the sample taken after symptoms reappeared [84]. One of the possibilities the authors suggested was that antibodies persisted the entire time, but they more strongly supported the hypothesis that IgM reappeared after reactivation of the acute infection. The data found by Lago et al. support the second hypothesis, since the group of congenitally infected children studied by them lost their IgM response before one year of age [30]. 

In conclusion, there is evidence that acute infection reactivation may induce IgM antibodies, although it is not a frequent phenomenon. Moreover, several of the findings described above may also be explained by reinfection (see below).

### 4.4. Reinfection

The research using animal models has long proved that concomitant sterile immunity against *T. gondii* is not achievable, i.e., the response that occurs against a first infection does not prevent the development of a challenge inoculation with a similar or different strain, although it usually develops a lower load [85,86]. Likewise, the complex life cycle and evasion mechanisms exerted by the parasite have precluded full vaccine successes because sterilizing immunity has not been raised [87,88]. However, is reinfection by *T. gondii* naturally frequent? The very low rate of vertical transmission to siblings observed years ago by Desmonts et al. suggested it is not [89]. However, this was an observation in regions of low prevalence, where chances of reinfection are rare. In fact, Gavinet et al. reported a reinfection in 1997: one pregnant woman with moderate titers of IgG, but no IgM, antibodies delivered a baby who presented ocular problems at nine months of age [90]. The serum samples from the mother and her son, as well as the pregnancy samples from the mother, were tested, revealing the appearance of IgM and IgA antibodies and an increase in IgG titers between 11 and 28 weeks of pregnancy.

Today it is rather common to find reports of cases with two or more strains in naturally occurring human or animal hosts exposed in highly endemic areas [48,91]. It is unclear whether these variants were acquired simultaneously or in subsequent infections; however, because secondary infections have been demonstrated or suspected [32,90], and it is relatively easy to induce reinfections -and even “superinfections”-in animal models [85,86], this cause of continuous induction of the immune response (and thereby of IgM antibodies) is feasible in highly endemic regions.

### 4.5. Natural Antibodies, “Innate” B Cells, Microbiota and IgM Autoantibodies

Natural antibodies comprise a possible important source of “chronic IgM”. These molecules were discovered a century ago (referred to in [92]). We recommend the interested reader to consult one of the highly qualified and extensive reviews available about this topic [92,93,94]. Here, we only summarize some important aspects. There are three subtypes of B lymphocytes: the best known are follicular/B-2 cells, responsible for the classical adaptive humoral response; after priming by antigens within the lymph nodes, some of them generate plasma cells (PCs) that produce IgM (primary response), and some others receive help from CD4+ Th cells and perform antibody class switch, differentiating thereafter into PCs that produce antibodies of other isotypes, of higher affinity and longer duration, or into memory lymphocytes, which migrate to different organs and respond to secondary stimuli. The other two subtypes are B-1 and Marginal Zone B cells (MZBs). B-1 lymphocytes originate and mature during fetal life, and after birth, they are located in the pleural and peritoneal cavities, where they are renewed, although they can migrate to the mucosae and respond there to stimuli [95]. MZBs are mostly derived from the bone marrow, like the B-2 cells, but rapidly respond to T independent antigens, especially from infectious agents. Both B-1 and MZB produce “natural antibodies”, which are important in protection against infections encountered early in life. B-1 and MZB cell-derived antibodies respond to BCR or TLR ligands like bacterial capsular polysaccharides or to self-molecules, such as phosphorylcholine; thus, they represent the first line of defense against many infections and help keep homeostasis by recognition and elimination of immune complexes, apoptotic cells and atherosclerotic plaques [96,97,98]. Besides the widely known IgM, natural antibodies of the IgA and IgG types are now recognized [93].

Newborns, pregnant women and healthy people have natural IgM and IgG that recognize *T. gondii* molecules and cross-react with other pathogens, the microbiota and even host antigens, such as HSP70 [10,99,100]. There is also evidence of natural IgM antibodies in swine and several other animals that activate the swine (but not felid) complement system against *T. gondii* tachyzoites [101].

The rheumatoid factor, a “natural” IgM (or IgG) antibody with autoreactivity against the Fc portion of IgG, was discovered long ago in newborns congenitally infected with *Treponema pallidum*, Cytomegalovirus and *T. gondii*, meaning that antibodies stimulated during gestation, apparently by a wide variety of pathogens (through TLR signaling or by reaction with polyspecific BCRs, can be autoreactive [102]. In fact, these molecules have been the concern in *T. gondii* infection diagnosis worldwide, and they are routinely destroyed or removed in order to improve specificity, although this does not always hasten the IgM decline [26,27,34,36,53,99,103,104].

In mice, B-1 lymphocytes are positively selected by self-antigens through BCRs, while the second activation and production of antibodies are regulated by pattern recognition receptors, such as TLR2 and TLR4 [105]. The TLRs are critical not only for antibody secretion but also for defining the steady-state B-1 immunoglobulin repertoire [105]. Profilins are small actin-binding surface proteins used by *T. gondii* to glide and spread on the external face of the host cell, indispensable for invasion [106]. These proteins induce murine B-1 synthesis of antibodies through TLR11 [107]. Another protein of importance in this regard is HSP70 from *T. gondii*, which may bind to TLRs, inducing proliferation of B cells [108]. HSP70 proteins are expressed in all living things -including parasites- and are highly immunogenic. Studies have shown that *T. gondii* HSP70 (Tg HSP70) is a B-1 cell, but not T cell, mitogen in infected mice, inducing anti-HSP70 autoantibodies, which were suggested as protective [109,110,111]. Humans do not express TLR11, but TLR2 and TLR4 ligation could induce natural antibodies by these or other parasite proteins, since they are activated in other, non-B cells such as monocytes upon infection [112].

Natural glycan-specific antibodies are partially produced by the microbiota-stimulated B-1 cells present in or migrating to the mucosa, and some of them could determine the response to pathogens [113]. It is worth mentioning that carbohydrate branches of glycosyl-inositol-phospholipids (GIPL) on the tachyzoite surface were reported as targets of *T. gondii*-specific IgM [103].

In a recent study with C57BL6 "bumble" mice, deficient in the transcription factor Nfkbid on which B-2 and B-1 cells depend for class switch and development, respectively, do not produce IgM antibodies and succumb to a secondary infection by *T. gondii.* If they are reconstituted with peritoneal B-1 cells from wildtype donors, they survive longer than non-transferred animals and produce IgM antibodies, which bind to tachyzoites and inhibit their invasion to 3T3 mouse fibroblasts (Fc receptor [FcR] negative) [114].

Important specific features of congenital toxoplasmosis are the route of infection (directly from umbilical cord blood), the stage of the infecting parasite (tachyzoite) and the maturity of the immune system [3,4,115]. These aspects may explain the difference between the kinetics of IgM antibodies in acquired/gestational and congenital infections, since a blood-borne systemic infection could target the lymph nodes directly, with less effect of the mucosal response and the microbiota. In these cases, Ig class switch would be more rapidly induced than in those that are acquired/gestational, challenged through the oral route.

### 4.6. Class-Switch Problems

Among the possible phenomena that underline the chronicity of IgM are the decrease in the number of B-2 or Th lymphocytes and the alteration of the transduction signals important for the Ig class switch.

There is evidence supporting both phenomena in toxoplasmosis, although we have not been able to find any studies that specifically address the effect on IgM levels derived from B-1, MZB or B-2 cells. When a host is infected, *T. gondii* tachyzoites spread throughout the body, and the number of T and B lymphocytes decreases by 50% in the blood and the bone marrow (BM); this was demonstrated in adult mice experimentally infected with the virulent RH strain [116]. Since the majority of B cells in blood and BM are B-2, much of the potential to produce secondary antibody classes would be diminished by these numerical alterations [96]. This becomes important because there is a decrease in lymphocyte counts in blood and BM, concomitant with an increase in the number of plasma cells in the latter. In agreement, a decrease in the number and function of naïve T cells exiting the thymus due to infection was also reported [117]; the authors suggested that systemic pathogens such as *T. gondii* may induce long-term immune disturbances associated with impaired thymic function.

Besides numeric declines, functional alterations may also explain the augmentation or permanence of IgM. It is worth remembering that two types of stimuli in B-2 and CD4+Th cells must occur for successful class-switching DNA recombination (CSR); these are BCR–antigen, CD40–CD40L (CD154) and TLR–ligand interactions, which directly or indirectly cross the PI3K route inhibiting it, to unblock the nuclear activating factor κB (NF-κB) and its translocation to the nucleus, where it provokes the synthesis of the activation-induced cytidine deaminase (AID, pivotal protein for CSR) and other genes related to this complex phenomenon [118,119,120]. Secondary stimuli delivered by T cells are cytokines such as IL-4, TGF-β, and IFN-γ, which activate signal transducer and activator of transcription (STAT) proteins in the B cells, as well as runt-related transcription factor proteins/RUNX, important for Ig-class-specific “S” transcripts that determine the Ig to be switched on. The interruption of the primary or the secondary stimulus causes the inhibition of CSR [118,119,120]. Chai et al. found that *T. gondii*-infected B and T cells had a smaller rough endoplasmic reticulum as well as a decreased level of cytokine secretion, respectively [121]. In a study carried out with infected macrophages, it was found that the parasite inhibits the response to IFN-γ through chromatin acetylation, thus blocking transcription [122]. This mechanism could be acting in lymphocytes. In their study, Chen et al. mention that excreted and secreted antigens (ESA) of *T. gondii* increased PI3K in EL4 cells, a tumor cell line used to explore T cell transcriptional regulation [123]. PI3K participates in the phosphorylation of fork-head box O transcription factors (FOXO), causing their degradation, and in this way inhibits AID expression [118,119,120].

CD40 is another primary stimulus that, together with the TLR, decreases the expression of the enzyme PI3K, in this way stimulating Protein kinase B (known as AKT) and FOXO, and favoring class switching. The effect of *T. gondii* on CD40 in B-2 lymphocytes is unknown, but an increase in CD40 expression has been seen in macrophages of C57BL/6 mice infected with type 2 but not type 1 or 3 strains [124]. It is known that an optimal range of CD40 stimulation is required for CSR stimulation (and somatic hypermutation) and that enhanced CD40 stimulation during a primary response triggers B cells to differentiate into plasmablasts rather than to proliferative or memory cells of the germinal center with Ig class switch [125].

Although the studies mentioned above were performed with macrophages or cell lines, these phenomena may be occurring in normal T or B lymphocytes, as they have similar receptors and signals that drive their functions. If true, the communication between T and B lymphocytes would be decreased, causing an inhibition of Ig class switch, either by alteration of primary or secondary stimuli or both (Figure 7).

## 5. Conclusions and Authors’ Viewpoints

The chronicity of the anti-*T. gondii* IgM antibodies can be confirmed from the data of several works, depicted together in this review. Despite some diagnostic tests, such as the ISAGA, making IgM look more chronic than others, such as IFAT, it seems that high titers prevail when the avidity is low, but the latter does not always reflect the tachyzoite menace to the host in terms of symptoms or vertical transmission.

We believe that there is more than one mechanism to explain the long-lasting IgM response, but some are less probable than others. For example, there is evidence that gestational sexual hormones can prolong the acute infection; nevertheless, this phenomenon is not exclusive of pregnant women; it also occurs in the acquired form of the disease and even in children with congenital infection. Likewise, reinfections could explain some but not most cases of persistent IgM, because they would provoke waves, not the slow decline observed. On the other hand, reactivation is not related to IgM, but to IgG raising, as in HIV patients or graft recipients.

Here, we propose various mechanisms that could better explain “chronic IgM”, although there is direct evidence of none. Among them, micro-reactivation (excystment)–encystment cycles of *T. gondii* could continuously induce IgM at low levels, but for long periods. Some strains could be more prone to provoke this stimulation, such as those of low virulence; the data from animals support this notion, although as far as we know, no long-lasting kinetic studies of IgM antibodies in experimental models have been performed.

Other mechanisms we think may cause IgM chronicity include B-1 cell-derived natural antibodies, since there is evidence of *T. gondii*-reactive Rheumatoid Factor in humans and IgM anti-HSP70 in rodents, the latter produced by B-1 cells. The recent work with “bumble mice” showed strong evidence of the role of B-1 cells, although this was proven in reinfections.

Ig class-switch alterations caused by the parasite or by altering receptors, cytokines or signaling molecules are likely, since interference with them by the parasite has been shown. Nevertheless, these results have been obtained with other cell lineages that are activated by the same molecules and means, but not in T or B cells directly.

It is clear that IgM anti-*T. gondii* antibodies decline slowly after infection. The biological and clinical relevance of this phenomenon is unclear, since it has been observed in asymptomatic pregnant women and congenitally infected babies, as well as in symptomatic cases of the acquired form. The underlying mechanisms are variable, and all or some may be occurring under different circumstances.

## Figures and Tables

**Figure 1 microorganisms-10-01659-f001:**
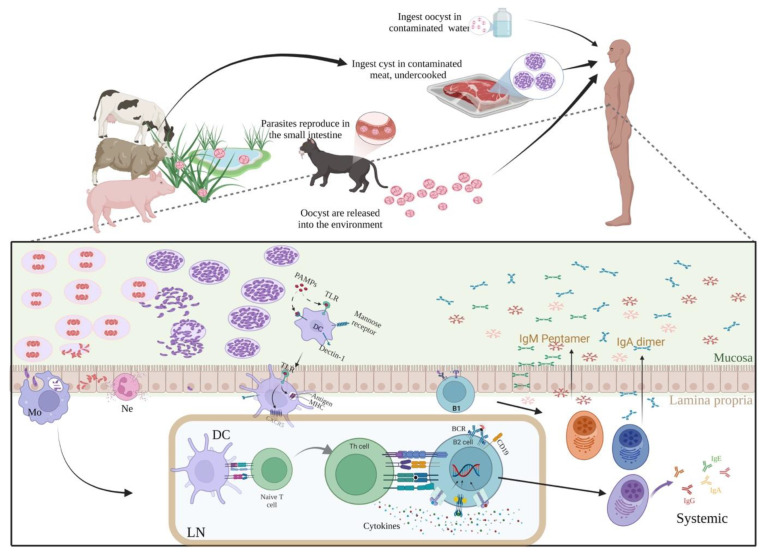
Immune response against *Toxoplasma gondii*. Both gestational and acquired toxoplasmosis occur by the ingestion of tissue cysts present in undercooked meat or oocysts that come from felines that release them with their feces; sometimes, water and legumes/pastures are contaminated. Thus, the parasite reaches the digestive tract. The sporozoite or bradyzoite released from the oocyst or tissue cyst, respectively, invades the enterocyte or crosses the epithelium to reach the submucosa, where there are gut-associated lymphoid tissues (GALT, such as Peyer’s patches) where a response begins: (a) Local macrophages (Mo) phagocytize the parasite and initiate an inflammatory response, together with local granulocytes (neutrophils, Ne); (b) B-1 cells can produce IgM and IgG antibodies (and IgA) after antigen or Pathogen-associated Molecular Patterns (PAMPs) recognition through the B cell Receptor (BCR), the Pattern Recognition Receptors (PRRs)- mainly Toll-Like Receptors (TLRs)- or both; B-1 and some marginal zone B cells (MZB) are able to spontaneously produce “natural” antibodies to microbiota, pathogens or autoantigens, that may cross-react with *T. gondii*; (c) Dendritic cells (DCs) take up antigens and travel to the nearest lymph node (LN) guided by chemokines recognized by specific receptors (such as CXCR5); at the LN, they stimulate both T helper lymphocytes (Th-CD4+), which convert into Th1 and produce Interferon-gamma (IFN-γ) and other cytokines, and cytotoxic T cells (CTL, CD8+), which become important for infected cell destruction; (d) The antigen also reaches the LN where the B-2 (follicular) B cells are primed and become available for CD4+ T help, undergoing antibody class switch. After selection of highly specific clones by DCs, they become plasma cells (PCs), which migrate to the bone marrow (BM) and secrete IgG, IgA or IgE antibodies. B-2, CD4+ and CD8+ memory cells are also born in the LN and migrate to different tissues, where they react more rapidly to a second stimulus. Protective mechanisms include complement fixation, opsonized phagocytosis, infected cell cytotoxicity by IFN-γ activated NKs and CTLs. This response is modulated by regulatory T lymphocytes and other cell types, including B-1 (B10) and M2 macrophages, directly or through tumor growth factor beta (TGF-β) or Interleukin (IL)-10. This summary is oversimplified to direct attention to antibody production. Please refer to relevant reviews of this topic for further details and other cell types and molecules involved [6,7,8,9,10,11,12]. Graphical contents were created with BioRender.com (license MX249HDDR2, accessed on 9 August 2022).

**Figure 2 microorganisms-10-01659-f002:**
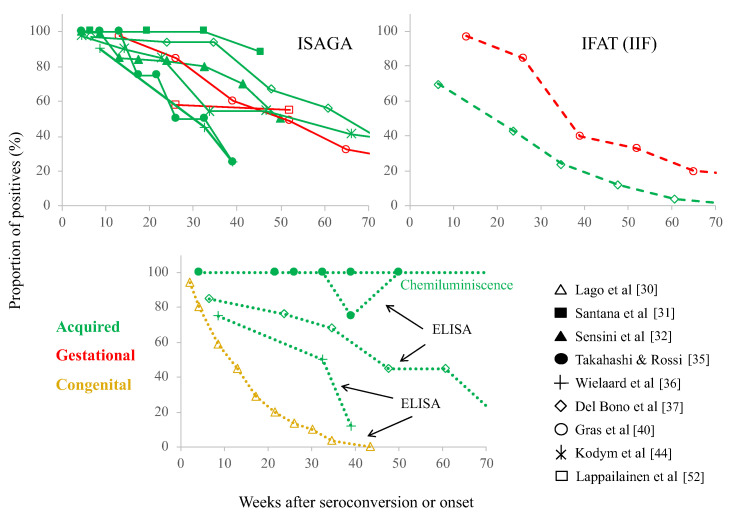
Proportion of cases positive for *T. gondii* reacting IgM antibodies over time in different groups after initial clinical or serological signs of infection [30,31,32,35,36,37,40,44,52]. Each graph corresponds to the test employed, as labelled. ISAGA: Immunosorbent Agglutination Assay; IFAT: Indirect Immunofluorescence Antibody Test (some articles refer to this technique as IIF or IFI); ELISA (EIA): Enzyme-linked Immunosorbent Assay; Chemiluminescence: microbead-based automated test. The abscissa axis cuts off at 70 weeks, but several studies followed the patients for longer periods.

**Figure 3 microorganisms-10-01659-f003:**
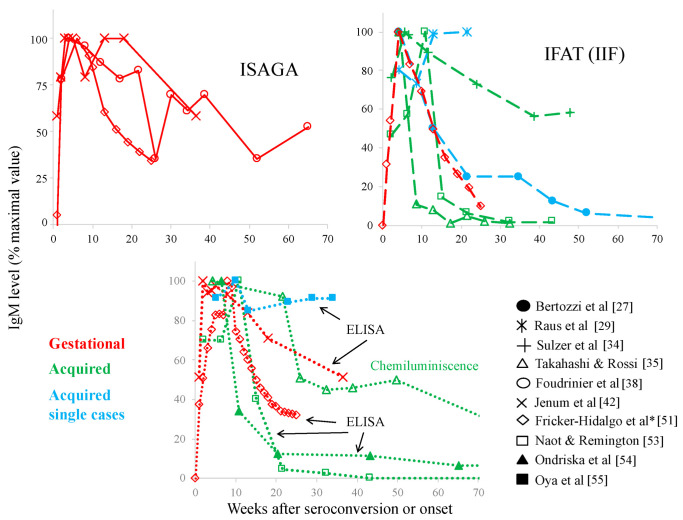
Kinetics of IgM antibody levels reactive to *T. gondii* in different groups after initial clinical or serological signs of infection [27,29,34,35,38,42,51,53,54,55]. The IgM levels were normalized to percentage of the maximum value in order to compare among studies. Dispersion data are not displayed (even if reported). The plots are separated according to the test performed. Acronyms of tests are as in Figure 2. The *x*-axis also cuts off at 70 weeks, but two studies followed patients for longer periods. * Fricker-Hidalgo et al. compared techniques, but we could only gather data from ISAGA, IFAT and ELISA.

**Figure 4 microorganisms-10-01659-f004:**
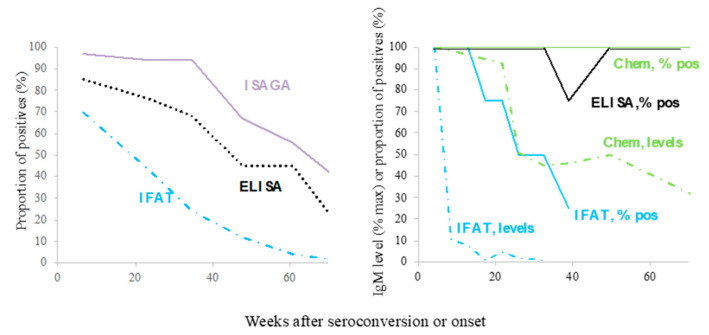
Kinetics of IgM decline assessed by different techniques and data presentation form (proportion of positives versus antibody levels) in two studies that compared these parameters with the same samples [35,37]. In both works, the patients had clinically active acquired infection.

**Figure 5 microorganisms-10-01659-f005:**
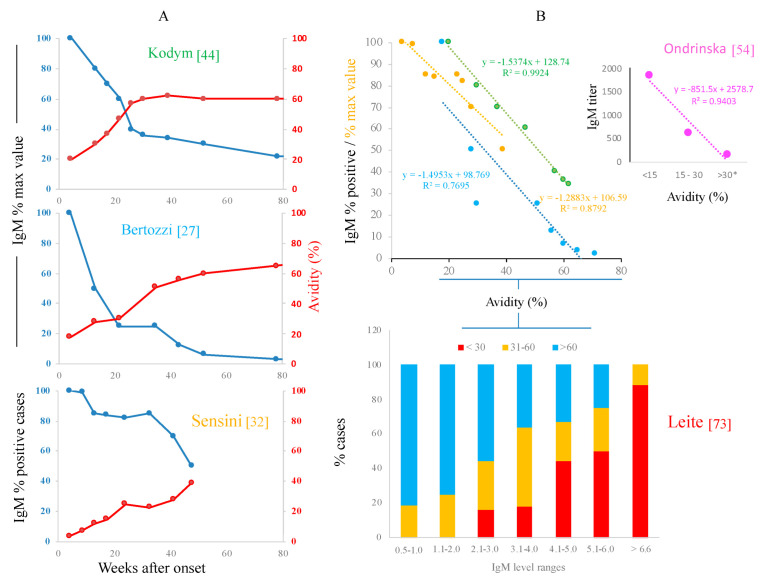
Relationship between IgM and avidity as seen in kinetics (**A**) or direct relation/proportion of cases (**B**). Data from tables or plots of the original works [27,32,44,54,73] were taken and used to build the graphs. Names in different colors are the first authors of these reports. * Considered chronic in that study.

**Figure 6 microorganisms-10-01659-f006:**
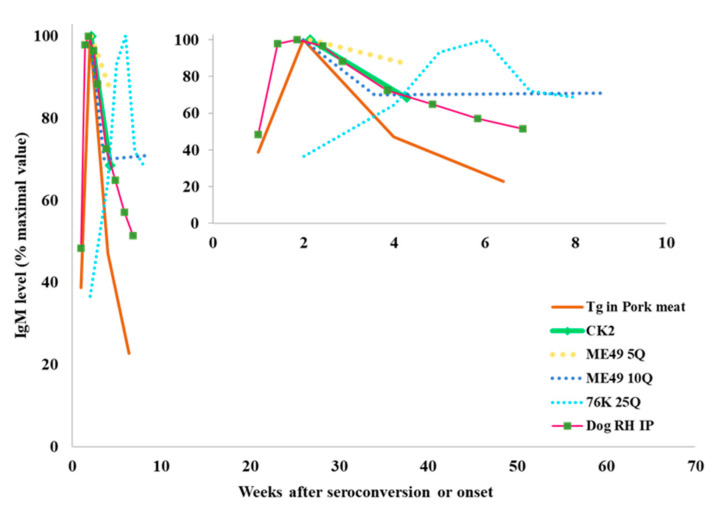
Kinetics of anti-*T. gondii-*reacting IgM antibody levels in animal groups experimentally infected. The large graphic shows the kinetics until 70 weeks in order to compare with human cases. The inset shows a shorter period amplified [75,76,77,78,79] and adjusted to percent of maximal value, with the purpose of normalizing and comparing among studies. CK2 and RH: virulent strains; ME49 and 76K: non-virulent strains; Tg: *T. gondii*; #Q: number of cysts inoculated; IP: intraperitoneal.

**Figure 7 microorganisms-10-01659-f007:**
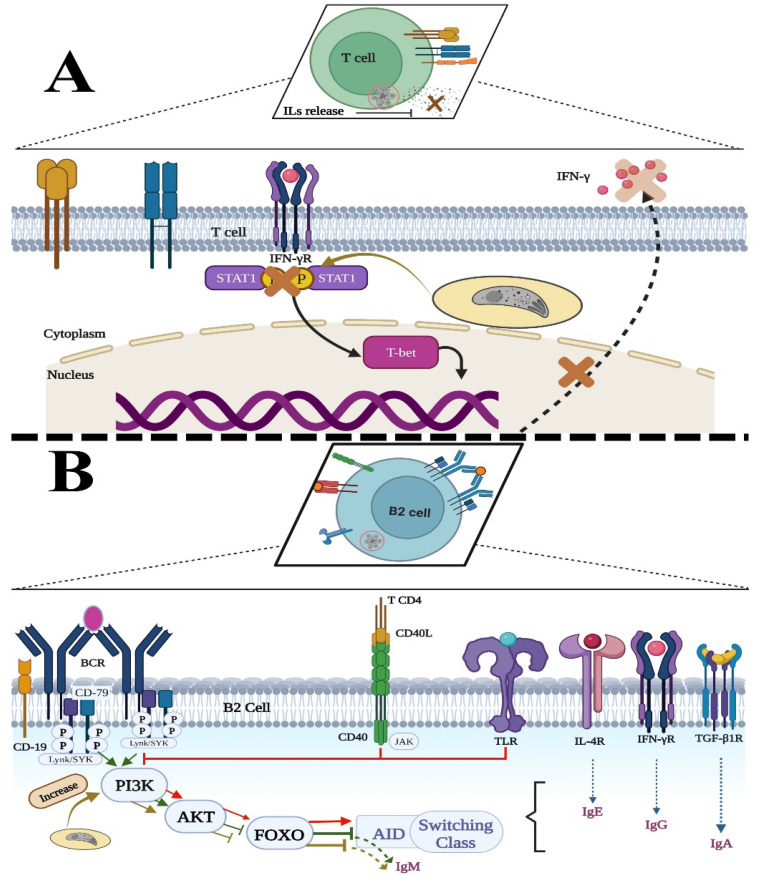
Possible mechanisms by which IgM can be prolonged: (**A**) Toxoplasma can infect CD4+ T lymphocytes. Within the cell, it could inhibit the phosphorylation of several receptors, resulting in minimal production of cytokines necessary for secondary stimulus in class switching. (**B**) Toxoplasma may invade B cells and enhances phosphatidyl inositol 3 kinase (PI3K) to levels that stimulate protein kinase B (AKT), inhibiting fork-head box O transcription factor (FOXO) and, consequently, the activation-induced cytidine deaminase (AID), a key enzyme for class-switch recombination (CSR). STAT-1: transducer and activator of transcription protein-1; T-bet: T cell transcription factor; BCR: B cell receptor; TLR: Toll-Like Receptor; IL-4R, IFN-γR and TGF-βR: Receptors for the cytokines IL-4, IFN-γ and TGF-β. Graphical contents were created with BioRender.com (licenses ZC249HCMVZ and PE249HCGKN, accessed on 9 August 2022).

## Data Availability

The data presented in Figure 2, Figure 3, Figure 4, Figure 5 and Figure 6 of this study are available in the Appendix A.

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
