# Peer review of "Anti-Toxoplasma gondii IgM Long Persistence: What Are the Underlying Mechanisms?"

_microorganisms, 2022, doi:10.3390/microorganisms10081659_

Round 1

Reviewer 1 Report

In this paper, Vargas-Villavicencio et al are questionning the mechanisms underlying anti-Toxoplasma gondii IgM persistence in different populations, especially from an immunological point-of-view.

It is a shame that it has not been reviewed by someone with expertise in toxoplasmosis diagnosis before submission, as they are several inaccuracies.

In addition, as the authors realized that their former hypothesis (IgM persistence limited to pregnant women) was unexact, which is not something new, the manuscript should have been structured differently and the biased hypothesis should not be mentioned.

Other comments:

Introduction :

Line 33 : Congenital toxoplasmosis does not result from transmission during lactation.

Line 89-90 : « we wondered about its occurrence in animals » : what for?

Figure 1 : In this figure, infected water appears as a central route of parasite transmission (bold characters, font size etc) which is not true, especially concerning transmission to humans.  In addition, there is no explanation of the upper part of the Figure in the footnotes.

Part2 :

Figures 3 and 4 : highly-sensitive techniques must be changed by the names of the techniques (ELISA). Indeed, regarding anti-Toxoplasma gondii IgM, ISAGA is considered as the most sensitive technique

Figure 4 : I suppose that, as for Figure 3, each curve corresponds to one study but legend is lacking

Figures 3 and 4 legends : IFAT is defined as Indirect Immunofluorescence Test. What does the A stand for ?

Part3 :

Line 127 :it seems to be in the right panel rather than in the left one

Lines 137-138 : must be enhanced in Figure4

Lines 161-164 : please rephrase

Lines 165 and 167 : Figure 4 : which curves ?

Lines 171-173 : I don’t believe that many studies report the uselessness of IgM for the diagnosis of acute infection. Indeed, in clinical pratice, IgM are routinely used and can help for the diagnosis of acute infection. Interpretation is based on antibodies titers and kinetics +/- on the results of additional techniques.

Part4

Line 242 pregnant women

Lines 260-261 : To date, only high avidity results can be interpreted and are indicative of a past infection. Low avidity results must still be considered as non-informative to date the infection, although some authors suggest that very low avidity results are highly suggestive of recent infections depending on the assay.

Line 324 : in 1997

Conclusion

Lin 458 : To date, avidity results still can not be used to confirm an acute infection (see above). Please rephrase

Line 478 : abs ?

All along the manuscript, rather use « acute infection » than « acute phase »

Reviewer 2 Report

Although the question in the title of the article was not answered, the manuscript presents a thorough review of the literature related to the persistence of anti-Toxoplasma IgM in the blood after overcoming the acute phase of infection. Standard or altered immune responses, stage conversion and virulence of Toxoplasma gondii strains, different patient groups, as well as variable specificities and sensitivities of serological tests are analysed as possible causes.

The overview article is written logically and comprehensibly. For a better understanding of the context of the received information, the authors supplemented the text with original graphs and diagrams. The reader will realize which aspects of toxoplasmic infection can contribute to the long-term persistence of IgM, but the authors do not make a definitive conclusion as to what is the cause of the investigated problem.

A few minor comments:

Line 42: “while IgE antibodies are unfrequently produced
against this parasite [10].”
this is not true, IgE is even used in diagnostics as a marker of acute toxoplasmosis. Figure 1- “lamina propia” - typing error should be propria Figure 2: Hypothesis visualization graphs can be confusing.
IgM can only extremely rarely fall on the cut-off level
3 months after infection. On the other hand, after reinfection
or reactivation (both extremely rare), immunoglobuline switch
is not repeated and rise of IgG, but not IgM can occur.

The reader will read all this on the following pages.
But if he does not notice that it is only a display
of hypotheses that are subsequently disproved, he may be
confused.
Line 119, Figure 3: Labels are confusing. “Highly sensitive
tests refer to ELISA (EIA) or fluorescent-chemiluminescent
immunosorbent (plate or microbead-based) techniques“.But in
which graph section are shown results of follow-up with these
techniques? At least one study, results of which are shown
in the section “ISAGA”, has been performed using ELISA IgM test.
Line 204 “Unfortunately, any of these studies followed
the changes in IgM and avidity values simultaneously.”
-
ambiguous wording.

The reviewer believes that the study may be of interest to all those involved in the immunology and diagnostics of toxoplasmosis.

Author Response

Please se attachment

Reviewer 3 Report

It is a very interesting paper about anti Toxoplasma IgM long persistence, which is a reality in routine serology. The authors have analyzed an important bibliography to summarize the available data and conclude that this phenomenon is common in many clinical situations. They tried to investigate the possible explanations. 

There is a problem in the numbering of the paragraphs, we have 4.5 (line 345), then 3.7 (line 415) then 5 (line 478).

Figure 1: all the the abbreviation arte not explained: BcR, PRRs, TLR ?

Line 114-115: the authors say that for the congenitally 114 infected children the decay was faster than most acquired cases; maybe the treatment could explain that.

In paragraph the authors speak about IgM titer: for IgM there are no international units, the presence of IgM is expressed as an index, and there no standardization between the different commercial kits. Line 139 the authors say Using Architect Toxo®, IgM titers were higher than those with the Vidas test. I don’t understand clearly what the authors want to say.

In the paragraph 4 the authors did not discuss about the fact that in infections by atypical strains, there is a long persistence of all isotypes (particularly IgM and IgA) with generally a very high index.

In the paragraphs 4.3 and 4.4 the authors discuss the problem of reactivation and re-infection. I agree with them to say that the presence of IgM is not a frequent phenomenon, but it is very common to detect specific IgA.
